# Perceptions of emergency care using a seizure care pathway for patients presenting to emergency departments in the North West of England following a seizure: a qualitative study

Leanne Rachel Male,[1] Adam Noble,[1] Darlene Ann Snape,[1] Peter Dixon,[1] Tony Marson[1,2]

¹Institute of Translational Medicine, University of Liverpool, Liverpool, UK
²The Walton Centre NHS Foundation Trust, Liverpool, UK

**Correspondence to**
Leanne Rachel Male;
hllmale@student.liv.ac.uk

## ABSTRACT

**Objectives** To establish the appropriateness of a previously developed seizure care pathway by exploring to what extent patients valued the intervention and perceived it as being helpful or not.

**Design** Qualitative descriptive study, using semistructured, in-depth interviews and thematic template analysis, theoretically informed by critical realism.

**Setting** In North West England, a seizure care pathway has been developed in collaboration with a specialist neurology hospital to support clinical management of seizure patients on initial presentation to the emergency department (ED), as well as access to follow-up services on discharge, with the aim of improving patient experience. Three National Health Service (NHS) EDs and a specialist neurology hospital provided the setting for participant recruitment to this study.

**Participants** 181 patients fulfilled the inclusion criterion with 27 participants taking part following their experience of an ED attendance and outpatient follow-up appointment after a seizure.

**Results** Five main themes emerged from the data: decision to seek care, responsiveness of services, waiting and efficiency, information and support, and care continuity. Two integrative themes spanned the whole study: lived experience and communication. This paper reports on two of the main themes: care continuity, and waiting and efficiency. The average time between ED presentation and interview completion was 100 days.

**Conclusions** Implementation of a care pathway is a complex intervention, requiring long-term follow-up to assess its integration into practice and effectiveness in service improvement. The seizure care pathway has the potential to enhance the care of seizure patients in the ED and at follow-up by improving continuity and management of care. The study demonstrates good aspects of the seizure care pathway as observed by patients and also recognises shortcomings within current service provision and questions what the NHS should and should not be delivering. Our study suggests various ways to enhance the pathway at service level to potentially drive improved patient experience.

## Strengths and limitations of this study

► This is the first in-depth study exploring patient experience of an emergency department (ED) seizure care pathway in the UK.

► The study allows the voices of both patients with known epilepsy and those experiencing their first seizure to be heard, as well as their carers/relatives, allowing for data to be collected from multiple viewpoints.

► Despite presenting at different National Health Service trusts, participants often reported similar experiences; issues may be more widespread, and it would be of interest to conduct further studies nationally.

► The study did not capture the perspectives of staff regarding the delivery of the care pathway in practice, which may give further insight into the appropriateness and effectiveness of its use.

► It remains to be established whether addressing the issues raised by participants about their experience of the pathway would translate into improved care quality and patient experience.

## INTRODUCTION

The National Health Service (NHS), like other health systems around the world, is operating within a context of rising demand, slow growth in funding and increasing operating costs. As a consequence, NHS providers ended 2015/2016 with a funding deficit of £2.45 billion.[1] The Five Year Forward View challenges the NHS to make unprecedented savings.[2] One way to achieve this is to reduce avoidable use of expensive emergency hospital services.

In the UK, seizures account for 3.3% of all calls to the emergency ambulance service,[3] ~60 000 seizure-related emergency department (ED) attendances and ~40 000 hospital admissions each year at

a cost to the NHS in 2015/2016 of ~£70 million.[4–6] Emergency care for seizures can be appropriate, important and even life saving. Despite this, evidence suggests that seizure patients' attendance at the ED is, more often than not, clinically unnecessary.[7–9]

Seizure-related ED visits are often expressions of suboptimal ambulatory care. The National Audit of Seizure management in Hospitals found that most of those presenting to the ED with known epilepsy (~65%) had not been seen an epilepsy specialist within the past 12 months but were in need of such support. They were often treated using suboptimal, outdated drug treatment regimens and so at risk of avoidable seizures and/or adverse events.[4] National Institute for Health and Care Excellence (NICE)[10] recommends when seizures are not controlled or treatment fails, the patient should be referred to a specialist for assessment. Evidence suggests such input could help improve patient outcomes.[11] ED visits do not, however, currently lead to most attendees receiving such specialist support.[4]

Part of the solution might lie with the introduction of a clinical care pathway within acute hospitals. Care pathways have previously been used effectively to improve care coordination in other conditions; however, there is limited evidence of their use in seizure care at present.[12] Importantly, a recent study in Ireland saw the implementation of a pathway specifically for seizures. It was designed to improve clinical management, reduce unnecessary admissions, minimise length of stay following ED attendance and improve follow-up services through a rapid access clinic.[13] Initial evidence suggests the pathway has utility in reducing length of stay for patients and time until the patient is followed-up by outpatient services.[13]

Ensuring equitable access to specialist neurological care in the UK has long been challenging. The above evidence suggests instituting a more proactive and better coordinated model of care may hold promise in driving up care quality while at the same time reducing emergency service use for seizures. Within the Cheshire and Mersey regions of England, a seizure care pathway was therefore developed. Seizure services in the area had been identified as a key target for improvement and better coordination.[4] The Cheshire and Merseyside Strategic Clinical Network coordinated the development of the pathway for adults presenting to emergency services with seizures in collaboration with stakeholders (neurologists, general practitioners (GPs), patient and public representatives and commissioners).

The pathway constitutes a paper pro forma that provides clinical advice on patient management and investigation, and instructions as to next steps and how to access the neurology/seizure service. Patients should be made aware of the seizure care pathway, offered an advice leaflet, be informed that they will be referred to the Walton Centre NHS Foundation Trust, which is the region's tertiary neuroscience hospital (and to expect to hear about an appointment in due course). Following patient ED assessment, the pro forma is faxed within 24 hours to the Walton Centre appointments office for processing. Patients will then be contacted by telephone to organise a neurology outpatient appointment in a time-appropriate manner, ideally within 2 weeks of their ED attendance, as per NICE guidance.[10]

The Care After Presenting with Seizures (CAPS) study, funded by the Collaboration for Leadership and Applied Health Research and Care, North West Coast, National Institute of Health Research (CLAHRC NWC), was set up to evaluate the pathway's effect. CAPS is a non-randomised trial comparing the effect of the pathway to usual care alone in reducing the number of emergency visits made by patients with seizures attending three local hospital EDs.

Government policy has, for a number of years, promoted the encouragement of patients to contribute to the planning and development of healthcare services as a means to providing them with a stronger voice in relation to decisions around their own health and care and to facilitate service development that reflects service-user needs and preferences.[2 14] The Five Year Forward View outlines the need for change in the relationship between people and the NHS, empowering patients to be actively involved in their own care, including working collaboratively on the development of health service delivery plans.[2] Such active patient involvement has been shown to improve outcomes and experiences for patients.[15] While the seizure care pathway has been designed with patient involvement, it remains important to continue this involvement following implementation of the pathway in clinical practice to help ensure it meets patients' needs, as well as clinical requirements.

This paper presents the findings of a qualitative study of patient's experiences of the seizure care pathway. Specifically, it aimed to explore: (1) whether the pathway was valued by patients and, if so, why?; and (2) what aspects of the intervention were perceived by patients as being helpful or unhelpful and why.

## METHODS
### Study design and setting
This work is a qualitative study conducted with semi-structured patient interviews. We were interested in a descriptive evaluation of the seizure care pathway based on participants' understanding, perceptions and experiences, both in relation to their ED care following a seizure presentation and the subsequent follow-up service offered.

### Participants
Table 1 outlines the inclusion/exclusion criteria for the study. Screening was undertaken by LRM, a PhD student with a nursing background.

Participants were recruited using purposive sampling, as detailed in the results section, to allow representation of the diverse range of patients attending the ED following a seizure.[16 17]

**Table 1** Participant inclusion/exclusion criteria for study

| Inclusion criteria | Exclusion criteria |
|---|---|
| ► Aged 16 years or older. <br> ► Presented in emergency department following a suspected seizure, regardless of whether they had a previous clinical diagnosis of epilepsy. | ► Provoked seizure (eg, alcohol withdrawal). <br> ► Event considered unlikely to be an seizure (eg, syncope). <br> ► Reside outside of Walton Centre's catchment area. <br> ► Learning disability. <br> ► Not fluent in written or spoken English. |

## Patient and public involvement

Patient involvement began at the outset during the design stages of the study, development of the topic guide and research questions. The project was supported by a local charity where public support was received and used in the form of recruiting pilot interviewees and support with topic guide development through discussion and sharing of ideas around patient preferences, priorities and previous experiences.

Results will be disseminated to participants through a lay summary of findings posted out to all participants following the analysis of all interviews.

## Data collection

Participant consent was obtained prior to commencement of the interview. Face-to-face interviews were conducted with the participant and, if requested by the participant, a carer—who had been present at the time of the ED attendance—was also invited to contribute. Interviews were audio recorded and transcribed verbatim.

The interview topic guide (box 1) was developed through a scoping review of the relevant literature.

**Box 1  Specific topics introduced within interviews**

Figure 1. Topic guide for interviews
► Can you tell me about what happened to make you attend emergency department (ED)?
  Probe: experience, decision making and transport to ED.
► Tell me what you remember about your experience in the ED.
  Probe: care processes, interactions with professionals, involvement in decision making, information and support provision.
► Was the most recent experience any different to past experiences of ED?
  Probe: similarities/differences, likes/dislikes and changes.
► What does 'good care' mean to you?
  Probe: expectations and shortfalls.
  What sort of changes, if any, could be made to improve your care throughout different stages in the ED?
► Tell me about your experience of the transition of care between ED and the Walton Centre?
  Probe: communication, timeliness and similarities/differences to previous referrals.
► Can you tell me about your experience of when you went for your outpatient clinic appointment?
  Probe: usefulness of appointment, benefits of pathway, improvement suggestions and how well do you feel your 'crisis' was dealt with?
► Discuss the usefulness of the current ED Questionnaire (Picker Institute) from patients' opinion.[50]

The topic guide was piloted, and iterative data analysis, researcher reflections and study team discussions were utilised to update the topic guide throughout the process of data collection.[18]

## Data analysis

Interviews were analysed thematically using the template method, informed by the work of King.[19] QSR International's NVivo V.10 software[20] was used to facilitate data management and document the analytical process. The analytic template was developed using a combination of a priori codes, derived from previous research around patient experience in general, and salient findings from early participant interviews. Using a priori codes ensured a focus on key areas potentially relevant to the study.[21] Further codes and themes were identified deductively through analysis of raw data, allowing identification of themes and patterns related to the study aims.[22]

The initial template was developed by LRM and another researcher (DAS). Transcripts were coded separately by LRM and DAS before comparison was made using the coding template.[23] Transcripts were read and reread; further amendments were made to the template until coding consensus was reached.[24 25] Final themes were reviewed by the study team to ensure they addressed the subject matter and accounted for the data content.

## FINDINGS

### Participant characteristics

Participants who met the inclusion criteria for CAPS, and had received a follow-up appointment following their ED visit, were invited to participate in the qualitative study. One hundred and eighty-one patients were screened for eligibility and contacted via telephone to invite them to take part in the study. Of these, 17 patients declined to participate. Reasons given for their refusal included anxiety, poor memory and time constraints. A further 137 potential participants could not be accessed due to patients not answering telephone calls/not replying to messages left and incorrect patient contact details.

The overall acceptance rate was therefore 22%, with 15% of those screened eventually participating. A total of 27 interviews were conducted between March 2016 and May 2017. Twenty-five of the interviews were conducted in the participant's own home, with the remaining two interviews taking place at an alternative location at the request of the participant. The average time between participants' ED presentations and interviews was 100

**Table 2** Patient demographics and clinical characteristics, grouped according to diagnosis following neurology review

| | Known epilepsy | First unprovoked seizure | Not epileptic seizure |
|---|---|---|---|
| | (n=13) | (n=8) | (n=6) |
| Age (years) | | | |
| Mean | 50 | 37 | 32 |
| Range | 24–78 | 17–64 | 19–45 |
| Gender, n (%) | | | |
| Male | 6 (46.2) | 5 (62.5) | 4 (66.7) |
| Female | 7 (53.8) | 3 (37.5) | 2 (33.3) |
| Employment, n (%) | | | |
| Full-time work | 3 (23.1) | 1 (12.5) | 2 (33.3) |
| Part time work | 2 (15.4) | 1 (12.5) | 0 (0) |
| Not employed | 8 (61.5) | 5 (62.5) | 3 (50) |
| Education | 0 (0) | 1 (12.5) | 1 (16.7) |
| Indices of Multiple Deprivation (IMD) regional decile, n (%) | | | |
| 1–3 (most deprived) | 9 (69.2) | 4 (50) | 3 (50) |
| 4–6 | 2 (15.4) | 3 (37.5) | 2 (33.3) |
| 7–10 (least deprived) | 2 (15.4) | 1 (12.5) | 1 (16.7) |

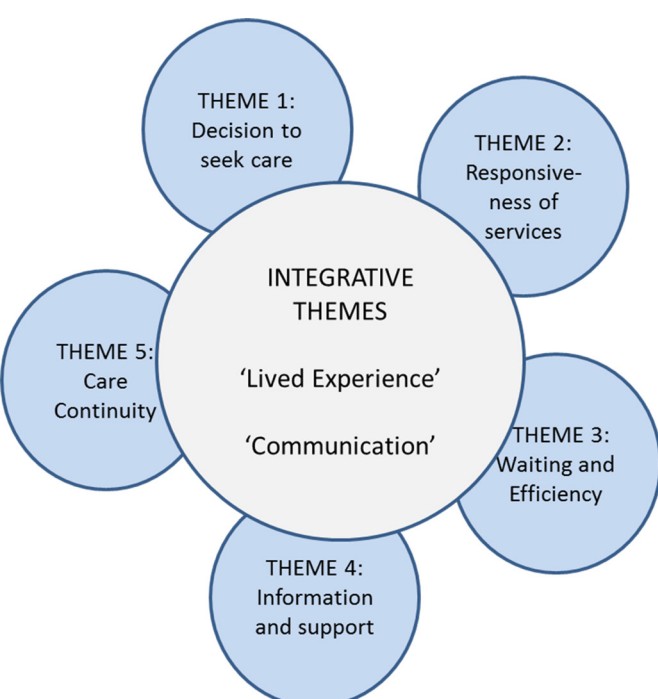

**Figure 1** Schematic diagram demonstrating the five main themes identified in the study, surrounding the two central and integrative themes of 'lived experience' and 'communication'.

days (SD 32.6). Demographics and clinical characteristics of the study sample are detailed in table 2. For clarity, 'Not epileptic Seizure' includes both non-epileptic attack disorder (NEAD) as well as other presentations referred on to neurology such as syncope.

### Identified themes

Analysis of the transcripts elicited five key themes (figure 1). In addition, two integrative concepts, 'lived experience' and 'communication', permeated participant discussions and as such were common across all themes.

The present paper focuses on findings from two of the five top level themes specifically due to the level of in depth description required to adequately explain such complex themes, namely, 'waiting and efficiency' and 'care continuity' as two of the most discussed themes within interviews. Care continuity, both within the ED and across services, was highlighted as the most important to participants in relation to their overall experience of the care pathway, along with waiting, which was frequently referred to throughout interviews. Each theme/subtheme will be discussed in turn in relation to participants' perception of the pathway as an effective intervention and examine each in relation to its perceived value to the pathway process overall.

### Waiting and efficiency

Time was viewed by most participants as a factor influencing experience. Irrespective of whether the participant perceived their treatment as being fast or slow, time

was continually mentioned, highlighting its importance in the process experienced by participants.

Participants' experience of waiting times varied at different points in their journey through the ED. Reference was repeatedly made to 'waiting for triage', 'waiting for tests' and 'waiting to see a doctor'. When participants perceived their wait time as being 'quick', particularly in comparison with previous experience, they reported positive feedback on the process.

[T]here was no question of she's going to wait or whatever; come right through and I say I saw that doctor… they were absolutely brilliant I couldn't fault them at all. P8, female, first unprovoked seizure

Conversely, participants perceived 'long' waiting contributed to poor experience.

[I]f your waiting on one of them beds to see a doctor… it does make you feel a bit on edge…people just get upset sitting out there. P29, male, known epilepsy

### Waiting and environmental resources

Participants frequently described the physical environment of the ED as 'busy'. A commonly reported feature was lack of bed space within the department, resulting in participants being left on trollies in the ED corridor for significant periods of time. As one female explained:

[T]hey were so backed up that we came in via ambulance and you know the way they have all the trollies down the corridor backed up, they were that bad that

the paramedics had to stay with me for 3 hours in a corridor. P5, kemale, known epilepsy

Similarly, this scenario had a significant impact on another participant's ED experience, prompting his self-discharge from the service. In this instance, the participant reported being in a postictal state but felt he no longer particularly required treatment; a decision informed, in part, by the belief he was a burden to the paramedics who had to remain with him during his time on a trolley in the corridor. He explained:

[T]here was the guilt trip I suppose on the last one because I had, this was adding to the anxiety because I was stuck on a trolley with paramedics looking after me and I'm thinking they could be out saving someone else's life. So that added to the pressure if that makes sense. Erm and that's why I said look you can't wait any longer and so I signed a self-discharge form because I felt guilty. P32, male, known epilepsy

Further interviews revealed participants' observations about the environment (often described as '*busy*' and '*noisy*') as well as their views about how it could be improved in a way that was more conducive to supporting those following a seizure. Suggestions for side rooms, dimmed lighting and a quieter area to enable patients to recover from their seizure in a calm and timely manner were articulated.

… they realised then they had to put me in a dark room to take my bloods and everything and then they said we'll try and put you somewhere quiet… they moved me up to the next waiting area… again there was the television on, flashing lights and we said you know you can't do that. If you put me in a dark room where it is nice and quiet my seizures will stop and calm down but they won't listen. P5, female, known epilepsy

### Time for care
Participants spoke about time in relation to waiting and with reference to time healthcare professionals (HCPs) spent engaging with them. While the majority of participants reported being satisfied with the time spent with HCPs, the visibility of staff members and the communication they received, others spoke of negative experiences related to the limited time spent specifically with HCPs.

Participants reported that waiting for a doctor to arrive often induced fear and uncertainty, triggering worrying thoughts about what might be wrong and what would happen next. This feeling was articulated mainly by those participants experiencing a first seizure, for whom the event and ED environment was largely unfamiliar. Overall, participants commonly held the view that doctors and nurses were '*in a rush*'. Such observations and experiences led to a sense of '*urgency*', felt by participants and a perceived inadequacy in support offered by HCPs.

I don't know how long the doctor was with me or the nurse but it didn't seem very long at all… yes I would have liked to have known more and the amount of time I spend in there, it seemed a very little time I spent with the doctor or the nurse. P12, male, known epilepsy

Those with known epilepsy tended to say they were more relaxed about the visibility of HCPs, not least because they expressed being more '*in control*' of the situation. More often than not, these participants reported a more positive experience of visibility of HCPs and were contented with the amount of time spent in consultation with doctors and nurses, even if this time was minimal and contact related mainly to routine basic observations.

[W]e were pretty much left to our own devices really, they checked that she was ok and then off they went. They weren't really regular check-ups because its more just waiting for her to come back round and be lucid and waiting for a doctor to turn up. P6, female, known epilepsy, carer

In contrast, a female with known epilepsy expressed great concern about the lack of contact with HCPs. Her concern led to fears for safety and a sense of heightened insecurity about the level of care received. These fears occasioned a negative perception of the care pathway and general experience within the ED.

[T]here was no medical staff available because they were so busy… I felt in danger because nobody knew what they were doing. P5, female, known epilepsy

### Conceptualisation of service provision
Despite acknowledging shortfalls in the service provided, often participants expressed understanding as to why this may be the case. Most commonly, limited staff and resources in the ED were highlighted as a potential cause of increased waiting times.

[I]t was just queue of trolleys with people in it. It was 4 and a half/5 hours just sat, lying around waiting… I didn't see anyone I was just lying there. P26, male, first unprovoked seizure

Participants also noted 'time of day' at which they attended the ED as a possible causal factor for increased wait, with certain times and days being recognised as particularly busy. These times included evenings, with particular reference to Friday and Saturday nights. At these times, participants said they believed EDs are regularly frequented by individuals who are intoxicated or who have injuries associated with alcohol and illicit drug use.

[I]t depends on what time of day you go, it depends on how you go, get taken into A&E it affects everything you know. P2, female, first unprovoked seizure

[I]f it was 10 o'clock on a Saturday night it would be totally different wouldn't it… P8, female, first unprovoked seizure

Often there were negative feelings expressed about the use of EDs in relation to alcohol and drug use. In particular the view that attendance under such circumstances represented service misuse.

> I think they were just happy to have a call out on a Friday night that wasn't a drunk in town basically. It was just an easy job for them. P6, female, known epilepsy

Finally, some participants recognised the use of the triage system, and the need to prioritise care of '*other, sicker patients*' as a cause for delays in their own treatment. Participants with seizures had typically been conveyed to ED via ambulance. This was seen by one participant as a quicker way into ED than the '*normal*' route of self-referral to ED and not attending by ambulance.

> [Y]es the normal way I think it is longer but with the ambulance taking you I think you get seen quicker. P1, female, known epilepsy

## CARE CONTINUITY
Three second-order themes were defined within the main theme of 'care continuity': perceptions of care continuity, accessibility of services responsive to need and sense of abandonment.

### Perceptions of continuity
According the pathway pro forma, the patient's attending doctor within the ED is responsible for informing the patient of the subsequent stages in their care pathway. This should be a timely referral for a neurology outpatient appointment. When asked about their understanding of this process, most participants acknowledged being told about the follow-up procedure, highlighting a positive example of communication within the ED.

In the few cases where participants were not told about the pathway, or when their waiting time for follow-up was delayed, this led to concern and anxiety, particularly among first seizure participants.

Conversely, those with known epilepsy expressed lower levels of anxiety in response to lack of advisement around follow-up procedures. Such participants often demonstrated their knowledge of the health service system, proactively accessing services they have previously used to obtain the follow-up care they wanted. This finding highlights that patients with known epilepsy, when provided with appropriate means, are able to proactively seek care to address their needs.

> [I]t's like we have our own pathway that just sort of made ourselves… but that sort of works because of the fact that for her condition it's easier to manage everything ourselves so we know what's going on. As far as the only pathway is when all the information gets handed over if she's got an appointment she wouldn't know what it's for. Whereas once she's

booked one directly with [epilepsy specialist nurse] she knows why she's going and what she wants to talk about. P6, female, known epilepsy

Epilepsy specialist nurses (ESNs) were only mentioned by participants with known epilepsy. In this group, participants spoke positively of the role and knowledge of nurse specialists. As one participant noted:

> [S]o [epilepsy specialist nurse] is absolutely brilliant… yes she's quite on the ball. P6, female, known epilepsy

However, those participants who were new to the service did not demonstrate any knowledge or understanding of the ESN role or related service provision.

### Accessibility of services to suit need
Participants experienced numerous barriers when transitioning from emergency care to follow-up ambulatory services. Important barriers were mainly organisational whereby participants frequently recounted experience of: difficulty in negotiating the appointment system, delays in receiving referral appointments, difficulty in accessing hospital/clinic locations and related transport problems.

Some participants noted that travelling to appointments was difficult. A combination of ill-health, inability to drive due to seizures, cost of taxis and poor public transport connections were noted as barriers to accessing services. This led to patients being reliant on family members to support them to get to their appointments, increasing a sense of burden and dependence in patients who were otherwise independent.

Where participants were expected to attend for more than one follow-up appointment, for example, where scans and further tests were required, they felt this compounded the level of inconvenience they experienced. Participants questioned the current process around 'follow-up' care provision. It was suggested, as a means to reducing the level of inconvenience experienced by patients, that appointments be combined to reduce the number of visits or, if possible, appointments be conducted in a local district general hospital suitable to the patient's location, thereby reducing the resources required for access.

> … because I can't drive now because of the epilepsy and I'm on my own with the children it's tricky, I've missed a couple of appointments at [tertiary clinic] just because it's tricky to get there. Because if I can't get anyone to give me a lift and look after the children it's hard for me to get there… it would be easier if they could come to my local doctors and do a clinic there. P35, female, known epilepsy

> I had to go back to [tertiary hospital] every day… and obviously from here it's like an hour and it was all that petrol so my dad had to take the week off work and take me back there every day before 9am which was tiring. P36, female, known epilepsy

## Sense of abandonment

Participants reporting of follow-up care after ED attendance varied significantly, demonstrating that all patients who had access to the pathway at ED did not necessarily receive the same benefits of the intervention post-ED discharge. Frequently, and irrespective of the level of information provided at ED discharge, this lack of continuity left patients feeling '*abandoned*'. This situation occurred predominantly at two time points: immediately after discharge from ED and following outpatient review at neurology clinic.

After discharge from ED, participants reported expecting to receive a follow-up appointment letter from the Walton Centre. In some instances, participants spoke of experiencing significant delays in receiving this appointment or not receiving an appointment. These experiences led participants to feel anxious, expressed as not knowing where to turn for further support. In addition, participants reported being faced with feelings of uncertainty about what to do and how long to wait before seeking further help.

> [I]t was just frustrating not knowing what was going on and then the fact I had to chase it up the appointment. If I hadn't chased it up three times I'd have never got the see the neurologist… I even went to the doctors [GP] and they [GP] left it up to me to chase them up and that is definitely wrong. P14, female, not epileptic seizure

The above quote reflects the actions of many participants within the study. Those having problems with hospital follow-up appointments often reported accessing their GP for assistance. This scenario, however, was seen to create further frustration with participants often feeling disappointed with GP care. Not least because it was felt the GP could not deliver the level of support needed by the participant and that clinical decisions were often referred back to neurology services.

Conversely, some participants reported feeling well supported by their GP with regard to their seizures. Although it was noted that GPs were often reluctant to make decisions with regard to antiepileptic drugs, the support and advice offered by GPs, through liaison with neurology services, had helped some participants feel aided to manage their condition without having to access emergency care.

> [S]he's [GP] been brilliant, she's been amazing, really, really good… I can't complain and you know if there was a problem I could just speak to the GP or I could ring someone at [Walton Centre] that's not a problem. I know there's always somebody available if you've got concerns. P35, female, known epilepsy

The second juncture when participants recounted feelings of abandonment was following their follow-up outpatient appointments. This more commonly occurred in patients who had experienced a first seizure, or a formal epilepsy diagnosis was not reached. These patients were typically discharged back to their GP after any necessary tests. When seizures continued participants were uncertain about what to do, often not seeing the GP as adequately trained to deal with such situations. This sense of abandonment was underpinned by beliefs that the service could not meet the needs of individuals. This led to frustration in relation to care, increasing anxiety and feelings of low mood within participants.

> I did take the letter in that the neurologist sent to me but all she [GP] kept doing was reading through it and typing what that letter says onto there's. She wasn't doing her own, I don't know, her own idea of what it could be or something I think she was just copying what the neurologist said because she didn't have… in my eyes she didn't have a clue in that doctors. P26, male, first unprovoked seizure

Participants repeatedly called for GPs to be better trained to deal with epilepsy and seizures. Participants believed this would enable GPs to provide appropriate, interim support while patients waited for outpatient appointments or during times between outpatient appointments and following discharge from neurology. In one particular case, a participant diagnosed with NEAD, lack of knowledge and understanding of the specific condition by the GP was felt to have hindered further follow-up care in the primary care setting resulting in a significant negative impact on her quality of life.

> I don't think she [GP] really knows what it is to be honest with you or how it affects you… it's such a new thing NEAD that they don't know what it is. Like I said I've never heard of it…if you had a seizure you had epilepsy basically that's what people think isn't it… I think there needs a lot more training in it because the way it seems, because they say it's in your head and you get people saying well if it's in your head you can get rid of it. P14, female, not epileptic seizure

This sense of abandonment and not knowing where to turn in time of 'crisis', on occasion, resulted in participants reporting overuse of the ED. Returning to the ED following each seizure, although recognised as not clinically necessary, was seen by some participants as a way back into the system. As one participant explained:

> I mean a lot of the time obviously again they email [tertiary hospital], they email your GP to say you've been there [A&E] you know and the doctor will look at that and just go right what's going on here why has she gone to A&E again and that's when they probably bring the appointment forward but yeah it makes you more inclined definitely to go to A&E… just to get seen and not have to wait weeks on end, months on end just for an appointment that's going to take 10 min… at least then even if you tell them what's going on at least they know what they're doing. P36, female, first unprovoked seizure

## DISCUSSION

To our knowledge, this is the first study to explore patient experiences and perceptions of an ED seizure care pathway. Despite variation in care experience throughout the process, some common themes were evident.

Waiting times and patient perceived efficiency within the service are fundamental elements of the care process. Wait times that were perceived to be significant contributed to poorer experiences of care. These findings are broadly consistent with other studies that demonstrate longer waits are associated with lower levels of patient satisfaction.[26–28] In the previous 15 years, emergency hospital attendances have increased by 47%, resulting in increased pressures on EDs.[29] Waiting in corridors was seen as an acceptable and somewhat 'standard' procedure to participants who had previous knowledge or experience within the ED.

Some participants described feelings of self-perceived burden as a result of their seizure attendance. This finding suggests that some patients in a postictal state may feel that their attendance at the ED is unnecessary and that they recognise they did not actually require the service. This is supported by a recent study conducted in a large hospital trust that concluded that the majority of people who have had seizures do not require the facilities of the ED for treatment.[7]

There is a need to realign our prehospital emergency care system to help reduce avoidable hospital attendances and admissions. Reducing the number of patients unnecessarily conveyed to hospital via ambulance through better training of ambulance staff may be a way to reduce ED burden.[30] Interventions to support self-management in patients have the potential to reduce hospital attendances, allowing patients to feel empowered in decision making surrounding their own healthcare needs.[31 32]

### Making time for care

Highlighted as key in this study is HCPs having the *time for care*. Time spent directly with patients has been a subject on the NHS agenda in recent years, with implementation of the NHS Institute for Innovation and Improvement's *Releasing Time to Care—Productive Ward* initiative in 2005, designed to improve patient satisfaction by increasing direct patient care by staff.[33] Positive results were seen as a consequence of the initiative in improvements in patient experience and in clinical and safety outcomes.[34 35] In our study, there were mixed experiences with regard to time spent directly with HCPs, with the majority reporting a positive experience, and few reporting negatively about their level of contact time. There is little evidence that the *Productive Ward* initiative[34] has been used within emergency medicine, and our findings suggest that this could be a useful tool to help improve patient experience.

### Discharge support and ongoing care

Poor care coordination resulted in the majority of participants reporting a *sense of abandonment* at some point in their journey along the care pathway. This included feeling unprepared to be discharged, lack of help and support from GPs and poor administrative communication with regard to follow-up appointments, all of which frequently left patients feeling anxious and helpless in their ongoing ambulatory care. This finding concurs with findings from the wider literature, with Hesselink *et al*[36] reported ED patients in general can feel unprepared at time of discharge. In our study, this finding demonstrates a shortfall in the effectiveness of the pathway process, highlighting potential for improvement to better support patients on discharge from the ED and their ongoing ambulatory care support.

A large proportion of participants felt the GP's support in their follow-up care to be deficient for their needs. Some GPs do not feel confident in dealing with epilepsy and making medical decisions resulting in treatment delays.[37 38] In these cases, as in our own study, patients have reported feeling frustrated and unhappy with care. Several suggestions to improve this aspect of care can be drawn from these findings. GPs have an integral role in care continuity of patients with long-term conditions.[39 40] Improvement in training of GPs in seizure management and epilepsy medications could reduce treatment delays and further ED attendances. Furthermore, GPs could take on added responsibility of providing a 'safety net' to try to help when communications break down between the ED and neurology services.

Further to the role of the GP, the ENS role is also designed to improve continuity and accessibility to neurology services.[41 42] The Walton Centre has been named an NHS Vanguard under the New Care Models scheme, which means they have national support to pioneer new ways of working that benefits patients.[43] As part of this scheme, 'integrated neurology nurses' have recently been employed to bridge the gap between the centre and the community, which aims to provide a better link between the hospital, the GP and community multidisciplinary teams.

Our findings support the suggestion of incorporating ongoing ambulatory care services, both from GPs and nurse specialists, within the care pathway. This would likely promote care quality across the continuum of the patient journey. Patients are faced with making difficult decisions when accessing care. Such decisions are based on their own understanding about what will be of most benefit and what will improve their care. Where patients feel at a loss regarding what to do next, a return to ED to prompt continuation of care has been reported as an option for some patients. There have been various studies around reasons for reattendance,[44 45] but there is little prior evidence showing that poor follow-up care results directly in reattendance to the ED. However, this study demonstrates that lack of adequate follow-up and *sense of abandonment* can result in some patients' reattending ED as a means to reaccessing the neurology system.

This present study also demonstrates that the entire seizure care pathway process needs addressing to ensure smoother patient transition and to allow patients to be

better supported in their ongoing care. We recognise both areas of good practice and areas where patients suggest improvements. These improvements, however large or small, need to be achievable in the context of the NHS. Raised patient expectations, medical and technological advances and an increased scope in what can be treated,[46] alongside a call to '*reshape the health services around the needs and aspirations of patients*',[47] have somewhat changed patient perception of what the NHS was originally set up to deliver in relation to demands now being placed on an already struggling service. The capacity of NHS service delivery needs to be taken into consideration when reviewing suggestions made for service improvement in order to make changes that will benefit patients and staff, as well as being both clinically and financially achievable.

### Strengths and limitations

Previous work in the field of seizure pathway evaluation did not qualitatively explore patient experience.[13] Our findings thus provide a more nuanced exploration of the pathway intervention, although sensitive to the context in which it is being delivered. Interviews were conducted in the patient's location of choice. This advantaged recruitment as the location was more convenient for participants and facilitated building rapport, as a non-clinical, more relaxed environment potentially encouraged participants to talk openly about their experiences. The Walton Centre catchment area is drawn from Merseyside, Cheshire, Lancashire, Greater Manchester, the Isle of Man and North Wales. The patient demographics are representative of the ED catchment area populations, where we would expect to see higher levels of unemployment and participants more likely to be living in the more deprived areas of the city.

Participants were recruited from three NHS study sites within NW England; all had a specific research nurse responsible for embedding the care pathway and follow-up procedure. This could potentially be a limitation, as the issues identified are specific to a nurse-led care pathway. In addition, this is a local initiative and as such findings from the current study may not be generalisable to other settings/geographical locations.

Added to this, participant uptake was low (ie, 22%). This figure may reflect a number of challenges associated with the conduct of this study including, for example, the chosen recruitment strategy, the difficulty in contacting patients by telephone and the nature of epilepsy as an illness. It is important to also acknowledge there is an inherent bias with studies of this nature; those opting to participate may have strong positive or negative feelings about the service which could potentially bias the results.

The average time latency of 100 days from ED attendance to interview may also have influenced the amount of information participants could remember about their experience. While this could not be avoided as participants had to have had their outpatient appointment before being interviewed, therefore experiencing the whole pathway, it should be recognised as a potential limitation to the findings.

This study did not capture perspectives of staff working within the field of the seizure care pathway. Previous quantitative work has though been conducted around staff perceptions of the barriers to implementation of a care pathway for seizure patients in Ireland[48] We recognise this as an important area likely to provide further insight around the care pathway implementation process in our study and are currently conducting a substudy to qualitatively explore the views of these individuals.

### CONCLUSIONS

Clinical, professional and organisational barriers all have impact on the delivery of complex interventions such as that of the seizure care pathway. That said, patients rely on such interventions, along with the support of HCPs, to support their transition from ED to ambulatory care. Although spanning a broad spectrum, there are various service-focused changes that have been suggested and discussed through our findings that may help to improve experience in this patient group. Refining interventions to support self-management, improved training for ambulance staff and more efficient use of current services such as ESNs and the 'integrated neurology nurses' could all prove useful in improving experience. It is crucial to the health service to take into consideration the suggestions made by patients in order to drive forward care based on what patients want. However, recognising the scope of the NHS, its values and the current financial challenges, it is important to recognise service improvements that will have the most impact. This will drive quality of care for service users and potentially support the NHS in reducing its current financial deficits.[49]

**Acknowledgements** The authors would like to thank the four NHS hospital trusts for their support with recruiting participants, patient advisers who supported the development of the study and those patients who gave their time to take part.

**Contributors** TM, PD and AN conceived the initial idea of the study, designed the trial and obtained research funding. LRM recruited, carried out and analysed the interviews, with input from DAS. LRM drafted the manuscript and PD, AN, DAS and TM made contributions to its development. All authors reviewed and approved the final manuscript.

**Funding** This work was supported by the National Institute of Health Research (CLAHRC NWC).

**Disclaimer** The views expressed are those of the author(s) and not necessarily those of the NHS, the NIHR or the Department of Health.

**Competing interests** None declared.

**Patient consent** Not required.

**Ethics approval** Ethical approvals were obtained from the research sponsor (University of Liverpool – protocol number: UoL001140) and from Wales Research Ethics Committee 7 (reference number: 15/WA/0207). Approval from HRA and the Research and Development department of each of the participating NHS hospital trusts (IRAS ID 173222) was also sought prior to commencing the study.

**Provenance and peer review** Not commissioned; externally peer reviewed.

**Data sharing statement** No unpublished data are currently available outside of the research study team.

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
