## [Reviewer comments · BMJ Open]

ARTICLE DETAILS

TITLE (PROVISIONAL)	Perceptions of emergency care using a seizure care pathway for patients presenting to emergency departments in the North West of England following a seizure; a qualitative study.
AUTHORS	Male, Leanne Rachel; Noble, Adam; Snape, Darlene Ann; Dixon, Peter; Marson, Tony

VERSION 1 – REVIEW

REVIEWER	Jennifer Williams Epilepsy Fellow Massachusetts General Hospital Boston USA
REVIEW RETURNED	08-Jan-2018

GENERAL COMMENTS	Abstract- no comments/changes Introduction  - Your introduction has some good details however it does not mention the general move towards patient centred care ie involving patients in the designing and planning of health care delivery which I think is actually the real essence of your work. I would mention this as a concept and relay how it relates to your project thus making it topical, unique and important. - I think a 1-2 sentences on the background to the pathway and how it works in your healthcare system, and if patients are aware they are placed on a pathway would be helpful. Methods Section  - Study Design and Setting- no comments to make - Participants- no comments - Data Collection  a) were the interviews taped or responses written - please clarify b) Also in lines 24-26 the authors mention " iteratively updated throughout the process of data collection, through ongoing analysis, researcher reflections and study team discussions"- does this mean that the interview structure was trialled prior to being put into full use in the study? ie was a pilot performed? c) Did you collect the time latency from ED presentation to interview? What was the average time that had elapsed- mention then in the discussion how this may affect your results Findings Section  -Participant Characteristics  a) did all patients agree to interview? How many refused and why. Was there difficulty with recruitment? b) Please comment on the catchment area and if the demographics of the interviewees matched that of the typical patients in your area- ie was your sample a fair reflection of your catchment population- I think it's important to mention this either here or in the discussion section under strengths or weaknesses c) Time from ED to interview not mentioned- this if possible should be stated- when did you decide to contact patients ie when , how
---

	and where did you recruit them? d) Also if you interviewed a patient who was supposed to have follow-up and you found that they did not what study procedures were in place to deal with this? Findings Section  - Perceptions of continuity a) you mention that there was "anxiety in response to lack of advisement around follow-up procedures"- I think the readers would be interested to know how the pathway ED to referral to specialist services link works- are there a number of steps- electronic or paper. Are the service contact details/ESN details given to the patient? This leads into the feeling of "lack of continuity left patients feeling abandoned" and "not receiving an appointment" Discussion Section  - It is a strength of your study that the interviews were performed in the patient's location of choice- you need to mention this in the strengths section of the discussion - With all studies of this kind there is perhaps an inherent bias to those who agree to interview ie those who would either have strong or negative feelings. I think you need to mention this as a source of bias. - The authors mention in the beginning of their strengths and weaknesses section that their study doesn't look at staff perspective of pathway integration into the ED work flow- perhaps mention the following paper if appropriate which looked at this issue could be included in your strengths and limitations section of your discussion " Moving evidence based guidelines for seizures into practice in the emergency department: What's stopping us?" Williams J1, Petrov G2, Kennedy U3, Halpenny J3, Doherty CP4. Epilepsy Behav. 2017 Jul;72:72-77. doi: 10.1016/j.yebeh.2017.04.022. Epub 2017 Jun 7.  - On page 19 (discussion section) The authors mention in lines 39-47 that there is a lack of papers evaluating the care pathway and representation to ED again if appropriate mention Seizure care in the emergency department. Identifying and bridging the gaps. A study of care and outcomes from 644 seizure presentations. Williams J1, Doherty J2, Di Blasi C3, Mabarak D4, Kennedy U5, Doherty CP6. Epilepsy Behav. 2018 Jan;78:226-231. doi: 10.1016/j.yebeh.2017.08.042. Epub 2017 Nov 10.  - Overall I felt that this was a very interesting topic on a very current and important subject- hammer that home in the conclusion - we need to address the findings found in your work if we are to strive towards patient centred care.
--	--

REVIEWER	Jon M Dickson The University of Sheffield, United Kingdom
REVIEW RETURNED	18-Jan-2018

GENERAL COMMENTS	My two main substantial criticisms/questions are: 1) Is this paper a process evaluation or a qualitative study of patients experiences? I think it's the latter. I would propose that the title, objectives and introduction are changed accordingly.
---

	2) The paper does not provide a sufficiently detailed description of the pathway itself. This should be added to the Introduction and/or the Methods.
--	---

REVIEWER	Arani Nitkunan Croydon University Hospital, London, UK
REVIEW RETURNED	22-Jan-2018

GENERAL COMMENTS	I would appreciate the authors addressing my following concerns:  1. Could the authors calculating the number of patients who would have presented during that timeframe and were evaluated via the seizure care pathway? This would enable readers to have a clearer understanding of what fraction of this population have been interviewed ie 27 out of ? 2. Most trusts use the Friends and Family test to assess patient satisfaction after an inpatient stay. Is there any way to incorporate results from this test as then it could be easily replicated by others as a quick way of assessing satisfaction. 3. Cultural differences may impact on communication. Could the authors comment on the ethnicity breakdown of their cohort of patients?
---

VERSION 1 – AUTHOR RESPONSE

Reviewer 1 Suggestions	
Your introduction has some good details however it does not mention the general move towards patient centred care ie involving patients in the designing and planning of health care delivery which I think is actually the real essence of your work. I would mention this as a concept and relay how it relates to your project thus making it topical, unique and important.	This comment has been noted and the introduction revised accordingly.
I think a 1-2 sentences on the background to the pathway and how it works in your healthcare system, and if patients are aware they are placed on a pathway would be helpful.	This comment has been taken into consideration and some background information about the pathway in general has now been added to the introduction section.
Data Collection  a) were the interviews taped or responses written - please clarify b) Also in lines 24-26 the authors mention " iteratively updated throughout the process of data collection, through ongoing analysis, researcher reflections and study team discussions"- does this mean that the interview structure was trialled prior to being put into full use in the study? ie was a pilot performed? c) Did you collect the time latency from ED presentation to interview? What was the average time that had elapsed- mention then in the discussion how this may affect your results 	 a) Interviews were taped- this has been clarified within the text. b) A pilot was performed- this has now also been made clearer within the article. c) This data was collected and has been presented within the data collection section as well as being discussed further in terms of potential limitations within the discussion.

Findings Section -Participant Characteristics a) did all patients agree to interview? How many refused and why. Was there difficulty with recruitment? b) Please comment on the catchment area and if the demographics of the interviewees matched that of the typical patients in your area- ie was your sample a fair reflection of your catchment population- I think it's important to mention this either here or in the discussion section under strengths or weaknesses c) Time from ED to interview not mentioned- this if possible should be stated- when did you decide to contact patients ie when, how and where did you recruit them? d) Also if you interviewed a patient who was supposed to have follow-up and you found that they did not what study procedures were in place to deal with this?	a) Recruitment strategy and participant agreement has been discussed within the findings and discussion sections. b) Demographics have also been discussed under strengths and weaknesses. c) Previously commented on within participant recruitment section and article has been edited accordingly. d) Patients were interviewed after they had already attended follow-up and so this did not appear to be a problem within the study. This has been clarified within the introduction section where the general working procedures of the pathway have been outlined.
Findings Section - Perceptions of continuity a) you mention that there was "anxiety in response to lack of advisement around follow-up procedures"- I think the readers would be interested to know how the pathway ED to referral to specialist services link works- are there a number of steps- electronic or paper. Are the service contact details/ESN details given to the patient? This leads into the feeling of "lack of continuity left patients feeling abandoned" and "not receiving an appointment"	This comment has been noted and the procedure involved is addressed within the general working procedures of the pathway section within the introduction.
Discussion Section - It is a strength of your study that the interviews were performed in the patient's location of choice- you need to mention this in the strengths section of the discussion - With all studies of this kind there is perhaps an inherent bias to those who agree to interview ie those who would either have strong or negative feelings. I think you need to mention this as a source of bias. - The authors mention in the beginning of their strengths and weaknesses section that their study doesn't look at staff perspective of pathway integration into the ED work flow- perhaps mention the following paper if appropriate which looked at this issue could be included in your strengths and limitations section of your discussion " Moving evidence based guidelines for seizures into practice in the emergency	These comments have all been noted and have been clarified/addressed within the discussion section.

department: What's stopping us?"	
Discussion Section  - It is a strength of your study that the interviews were performed in the patient's location of choice- you need to mention this in the strengths section of the discussion - With all studies of this kind there is perhaps an inherent bias to those who agree to interview ie those who would either have strong or negative feelings. I think you need to mention this as a source of bias. - The authors mention in the beginning of their strengths and weaknesses section that their study doesn't look at staff perspective of pathway integration into the ED work flow- perhaps mention the following paper if appropriate which looked at this issue could be included in your strengths and limitations section of your discussion " Moving evidence based guidelines for seizures into practice in the emergency department: What's stopping us?"	Figures and numbers have been adjusted accordingly with more up-to-date searches.
Reviewer 2 Suggestions	
1) Is this paper a process evaluation or a qualitative study of patients' experiences? I think it's the latter. I would propose that the title, objectives and introduction are changed accordingly.	This comment has been taken into consideration. The whole project as a PhD is a process evaluation but I agree that this specific article focuses primarily on the qualitative aspect of patients' experiences. As a result, the title, objectives and introduction have been edited accordingly to focus the article specifically to the qualitative study.
2) The paper does not provide a sufficiently detailed description of the pathway itself. This should be added to the Introduction and/or the Methods.	This comment was similar to one made by Reviewer 1 and subsequently this has been addressed within the introduction section.
Reviewer 3 Suggestions	
1. Could the authors calculating the number of patients who would have presented during that timeframe and were evaluated via the seizure care pathway? This would enable readers to have a clearer understanding of what fraction of this population have been interviewed ie 27 out of?	This comment is similar to one made previously by Reviewer 1 about recruitment. This has been taken into consideration and this information has now been added to the article accordingly.
2. Most trusts use the Friends and Family test to assess patient satisfaction after an inpatient stay. Is there any way to incorporate results from this test as then it could be easily replicated by others as a quick way of	We are addressing measures of experience rather than satisfaction and therefore we would ask slightly different questions with regard to whether or not certain aspects of care happened (i.e. how long people waited, where they given pain relief if

assessing satisfaction.	required?) rather than satisfaction type questions (i.e. how likely are you to recommend...etc?) There is potential that in the future such measures could be used in parallel although the current research does not focus on satisfaction and therefore this would require some further investigation going forward.
3. Cultural differences may impact on communication. Could the authors comment on the ethnicity breakdown of their cohort of patients?	This is a very interesting point made by the reviewer regarding cultural differences and how this may impact communication within the ED and beyond. Unfortunately, as part of this research we did not collect information about participants ethnicity and so unfortunately cannot provide a breakdown of their cohort. In any future research around the subject I would consider collecting this information as I do believe it to be important and it may have been an oversight and potential limitation within this project.

VERSION 2 – REVIEW

REVIEWER	Jennifer Williams Academic Unit of Neurology Trinity Biomedical Science Institute Pearse Street Dublin Department of Neurology Massachusetts General Hospital 55 Fruit Street Boston MA USA
REVIEW RETURNED	11-Apr-2018
GENERAL COMMENTS	The comments have been well addressed by the authors. This is an important body of work as we try to move toward a model of patient centred care.
REVIEWER	Jon M Dickson The University of Sheffield, UK.
REVIEW RETURNED	08-Jun-2018
GENERAL COMMENTS	The reviewer provided a marked copy with additional comments. Please contact the publisher for full details.
REVIEWER	Arani Nitkunan Croydon University Hospital 530 London Road CR7 7YE United Kingdom
REVIEW RETURNED	07-May-2018
GENERAL COMMENTS	I am grateful to the reviewers for addressing our queries. I would suggest that the data of 181 patients who fulfilled the criteria for inclusion in the study is mentioned in the Abstract as well as including the average time taken between ED presentation and interview of 100 days.

VERSION 2 – AUTHOR RESPONSE

Reviewer 2 Suggestions	
Revision of strengths/limitations	Comment has been taken into consideration and text has been edited accordingly
Comment on page 4, line 43/44	Minor revision to text made accordingly
Comment around wording in Table 1	The word ‘unprovoked’ has been removed
Comment on page 7, line 20	‘NHS’ added here to clarify
Comment on page 7, line 32	Clarified in text that a scoping review was conducted as opposed to a systematic review
Comment on Table 3 regarding what is meant by ‘Not epileptic seizure’ in the table	For clarity, the description of this category has been added to the text above. ‘Not epileptic seizure’ includes both NEAD patients as well as those referred onto neurology who had not had a seizure, i.e. syncope
Comments on page 10- 1. Statement that there are 5 themes and 2 integrative themes. The writer states they will describe themes 3 and 5 without explanation of why they will not describe the other 3 themes.	These two themes were chosen specifically due to the importance they were given by participants within the interviews. Also, due to limitations with word count etc, it would not have been possible to discuss all 5 themes in adequate depth within the word limit so two themes were also selected for this reason.
Comments on page 10- 2. The headings of the results largely don’t correlate with the themes in Figure 1. Why is this?	The themes in Figure 1 are the 5 overarching themes. What has been described within the paper are the subthemes using subtheme headings within two of the themes; ‘Waiting and efficiency’ and ‘Care Continuity’. Figure 1 does not include subthemes as this would be too complex to image within one figure and would be overly confusing to readers
Interchangeable use ‘care continuity’ and ‘continuity of care’ needs to be addressed to use one term throughout	This has been addressed and ‘care continuity’ has now been used throughout
Reviewer 3 Suggestions	
I would suggest that the data of 181 patients who fulfilled the criteria for inclusion in the study is mentioned in the Abstract as well as including the average time taken between ED presentation and interview of 100 days	This comment has been considered and subsequently address